# Preparation and Characterization of Biomass Tannin-Based Flexible Foam Insoles for Athletes

**DOI:** 10.3390/polym15163480

**Published:** 2023-08-20

**Authors:** Zhikai Zuo, Bowen Liu, Hisham Essawy, Zhigang Huang, Jun Tang, Zhe Miao, Fei Chen, Jun Zhang

**Affiliations:** 1Physical Education Institute, Southwest Forestry University, Kunming 650224, China; zuozhikai@swfu.ed.cn (Z.Z.); hzg123hzg@163.com (Z.H.); chenfei520123@163.com (F.C.); 2Yunnan Provincial Key Laboratory of Wood Adhesives and Glued Products, Southwest Forestry University, Kunming 650224, China; 3Department of Polymers and Pigments, National Research Centre, Dokki, Cairo 12622, Egypt; hishamessawy@yahoo.com; 4Yunnan University, Kunming 650500, China; 20220038@ynu.edu.cn; 5Yunnan Arts University, Kunming 650000, China; mz910211@sina.com

**Keywords:** tannin, furfuryl alcohol, polyvinyl alcohol, foam, insole

## Abstract

The exploitation of bio-based foams implies an increase in the use of renewable biological resources to reduce the rapid consumption of petroleum-derived resources. Both tannins and furfuryl alcohol are derived from forestry resources and are, therefore, considered attractive precursors for the preparation of tannin–furanic foams. In addition, toughening modification of tannin–furanic foams using polyvinyl alcohol (PVOH) results in a more flexible network-like structure, which imparts excellent flexibility to the foams, whose relative properties are even close to those of polyurethane foams, which are the most used for fabrication of insoles for athletes. In addition, the addition of PVOH does not affect the thermal insulation properties of the foams by testing the thermal conductivity, resilience, and elongation at break, while reducing the brittleness of the samples and improving the mechanical properties. Also, the observation of the morphology of the foam shows that the compatibility between PVOH and tannin–furanic resin is good, and the cured foam does not show fragmentation and collapse, while the bubble pore structure is uniform. The developed flexible foam derived from biomass resources endows the foam with good thermal insulation properties and high mechanical properties, and the samples exhibit suitable physical parameters to be used as flexible insoles for athletes.

## 1. Introduction

As a necessity in everyday life, footwear serves to protect the feet from injury or irritation and also has an aesthetically pleasing and comfortable design that can be matched to a variety of outfits and occasions. According to a report by the APICCAPS, global footwear production totaled 22 billion pairs in 2021, an increase of 8.6% from 2020, and is expected to grow every year [1]. Since the increasing consumption of footwear will result in a large amount of polymer waste in our environment, it is essential to develop footwear components with biodegradable properties [2,3]. Insoles are a major footwear component that enhances the wearer’s comfort. Especially for athletes during exercise, soft insoles with shock absorption can effectively protect their feet. Insoles can be classified as foam insoles [4], rubber insoles [5], and ethylene–vinyl acetate copolymer (EVA) insoles [6], and foam insoles are widely favored for their good shock absorption, breathability, and anti-slip properties [7]. The main raw materials for preparing traditional foam insoles come from petroleum derivatives, such as polyurethane (PU) insoles, polyvinyl chloride (PVC) insoles, and EVA insoles. Due to the large consumption of petrochemical resources, the rising prices of petrochemical raw materials, and the release of large amounts of carbon dioxide during processing, it is important to explore renewable materials to replace petroleum-derived materials in the preparation of insoles for athletes.

In recent years, natural biomass materials have been used to substitute some petroleum-based materials to prepare bio-based materials, such as biomass adhesives [8,9,10], foams [11,12,13], plastics [14], and grinding wheels [15]. For biomass foams, condensed tannin–furfuryl alcohol co-condensation resin-based foams have been mostly studied [16,17].

Tannins, mainly extracted from bark, are the most abundant natural aromatic biomolecules in nature [18]. Cross-linking between tannins and furfuryl alcohol has been demonstrated, while furfuryl alcohol is derived from the processing residues of agricultural and forestry crops, again a widely sourced biomass material. Under acidic conditions, tannins and furfuryl alcohol can undergo co-condensation reactions to obtain tannin–furan (TF) foams. This foam has attractive properties such as a light weight, excellent refractoriness, and thermal insulation [19,20,21]. However, the self-condensation reaction of furfuryl alcohol occurs under acidic conditions, which dominates the reaction, and almost no co-condensation reaction with tannins occurs [22]. In addition, furfuryl alcohol has no significant reactivity under alkaline conditions, whereas the self-condensation and hydroxymethylation reactions occur mainly under acidic conditions [23]. To enhance the reactivity of tannins and furfuryl alcohol, formaldehyde has been used as a cross-linking agent and helps to obtain tannin–furfuryl alcohol–formaldehyde (TFF) foams with good mechanical properties [12]. However, the toxicity and health hazards of free formaldehyde have limited its further development. A previous study showed that glyoxal can completely replace formaldehyde and can be used as a cross-linker for the reaction with furfuryl alcohol and tannin under acidic conditions [24]. Ether, as a low-boiling liquid that can be volatilized at room temperature, is a blowing agent for the preparation of foams, and the foams can be prepared quickly without heating. However, in this case, the foaming process is difficult to control, resulting in an irregular distribution of pores in the foam, which is not favorable for applications. In our previous study [25], we suggested that the addition of Tween 80 as an emulsifier to the tannin–furan resin system could change the interfacial tension of the molten polymer, resulting in bubble wrapping, which is essential for the preparation of foams with uniform pore distribution. The lower hardness and compressive strength of the foams prepared with glyoxal as a cross-linking agent compared to formaldehyde was attributed to the fact that the intercellular polymerization was not rapid enough, which led to bubble coalescence during foam blowing. This makes the foam material show greater brittleness and a higher degree of pulverization. Therefore, toughening modification of the brittle material is needed to improve the flexibility of the foam material.

In the study of toughening modification of materials, the addition of a toughening agent is usually used. Zhou et al. [26] toughened and modified the epoxy resin using a rubber copolymer, which improved the tensile and flexural properties, but the heat resistance of the system was reduced. There is a study that successfully prepared shape memory phenolic resins by introducing different contents of polyurethane prepolymers into phenolic resins [27]. With the increase of polyurethane prepolymer content, the elongation of the modified phenolic resin gradually increased, and the cured phenolic resin foam had a certain degree of toughness. Polyvinyl alcohol (PVOH) is an excellent performance toughening modifier with good water resistance, chemical and thermal stability, and it is biodegradable in the environment. PVOH molecular chains contain a large number of hydroxyl groups and hydrogen bonding, which can be cross-linked with the natural polymer material to form a dense network structure, thus, enhancing the mechanical properties of the composite material [28]. Therefore, PVOH is widely used as a toughening modification of resin-based materials, such as foams [29], films [30], and hydrogels [31].

Therefore, based on the above facts, the PVOH as a toughening agent was used to improve the tear strength, tensile strength, and elongation at break of the tannin–furanic resin-based foam. We hope that the prepared flexible biomass foam insoles can expand the application of tannin in industry and provide technical reference and theoretical guidance for the development and application of new bio-based insoles for athletes.

## 2. Materials and Methods

### 2.1. Materials

Bayberry (Myrica rubra) tannin powder (T, 85%) was purchased from Shengxuan Chemical Company (Zhengzhou, China). Furfuryl alcohol (F, 98%), glyoxal (G, 40%), p-toluenesulfonic acid (pTSA, 97.5%), ether (99.5%), and PVOH (5%) were obtained from Sinopharm (Beijing, China). Tween 80 and guar gum (viscosity of 5000–5500 mPa·s) were purchased from Shandong Kepler Biotechnology Co., Ltd. (Jining, Shandong, China).

### 2.2. Preparation of Various Flexible Tannin–Furan Resin-Based Foams

According to the formula in Table 1, the tannin powder, furfuryl alcohol, and glyoxal were added into a 250 mL beaker and stirred evenly to obtain tannin–furfuryl alcohol–glyoxal (TFG) resin [16]. Then, Tween 80 as the emulsifier, guar gum as the release agent, and PVOH as the toughening agent were added and stirred with an egg beater at a low speed (50 r/min) for 10 min. Finally, the p-TSA (30% aqueous solution) and ether as foaming agent were added to TFG resin under stirring for 5 s to obtain the TFG-PVOH (TFGP) mixture, and the mixture was placed in an oven at 60 °C for 30 min to foam evenly. Then, the sample was taken out to obtain the flexible TFGP foam. Its preparation process and appearance are shown in Figure 1.

T resin was obtained by adding 20 g of tannin and 10 g of distilled water to a beaker and mixing well. Then 4 g of Tween 80 and 10 g of guar gum were added and stirred with an egg beater at low speed (50 r/min). The mixture was placed in an oven at 60 °C for 30 min to allow for uniform foaming to obtain T foam. At the same time, 15 g of PVOH was added to the mixture and stirred with a whisk at the same speed to obtain the TP resin. The TP resin was foamed in the oven under the same conditions to obtain TP foam.

### 2.3. Characterizations

The structure of each foam was investigated using a Varian 1000 infrared spectrometer (Varian, Palo Alto, CA, USA) and ESI-MS spectrometer (Waters, Milford, MA, USA). In the FTIR examination, the sample’s preparation involved mixing 1 g of KBr with 0.01 g of each sample in powder form, and the investigation run covered a wave number range of 500 to 4000 cm^−1^.

The foam samples were cut into squares with dimensions of 5 mm × 5 mm × 5 mm, and the surface of the samples was sprayed with gold. Foam samples were observed and photographed in a Hitachi, S-4160FE field emission scanning electron microscope (Hitachi, Tokyo, Japan).

The apparent density of foam samples was measured according to the national standard GB/T 6342-2009. The size of samples was measured, and Equation (1) was applied to calculate the apparent density:(1)$$\rho =\frac{m}{v}\times 10^{6}$$
where *m* is the mass of the sample, g; *v* is the volume of the sample, mm^3^; *ρ* is the apparent density, kg/m^3^.

Thermal conductivity can effectively measure the insulation performance of the sample, and a lower thermal conductivity can maintain the foot temperature and improve the comfort of the wearer. The thermal conductivity of the samples was tested using a thermal conductivity meter (YBF-2, MINSKS, Xian, China). The samples were cut into cylinders with a radius (R) of 50 mm and thickness (h) of 10 mm, and the tested data was processed according to Equation (2):(2)$$\lambda =-mc\frac{2h_{p}+R_{p}}{2h_{p}+2R_{p}}\cdot \frac{1}{\pi R^{2}}\cdot \frac{h}{T_{1}-T_{2}}\cdot \frac{dT}{dt}|T=T_{2}$$
where *λ* is the thermal conductivity of the foam, W·m^−1^·K^−1^; *m* is the mass of the lower copper plate, g; *c* is the specific heat capacity of the copper block, $R_{p}$ and $h_{p}$ are the radius and thickness of the lower copper plate, mm, respectively; *R* is the radius of the sample, mm; *h* is the height of the sample, mm; $T_{1}-T_{2}$ is the temperature difference between the upper and lower copper plates; $\frac{dT}{dt}|T=T_{2}$ is the cooling rate of the copper plate exposed to the air.

The Shore hardness tester is often used for soft plastics and rubber materials. The test method is to press a specific indenter into the material under specific conditions and measure the hardness of the material by the numerical value displayed by the hardness tester. In this experiment, a Shore hardness tester (LX-C, HANDPI, Shanghai, China) was selected to test the hardness of foam samples. Hardness is the main feature of any outsole, midsole, or foam insole because it is related to flexibility and cushioning effect.

The rebound rate of the samples was tested according to the national standard GB/T 6670-2008. The samples were cut according to the dimensions (100 ± 0.5) mm × (100 ± 0.5) mm × (50 ± 0.5) mm, and then tested using a falling ball rebound meter (PMLQ-400, Beiguang Precision Instrument Co., Beijing, China). The drop height of the steel ball was 500 mm ± 0.5 mm and the mass of the steel ball was 16.8 ± 1.5 g. Three results were obtained for each specimen, and the average value was taken.

Under the environmental conditions of room temperature 23 °C and relative humidity not more than 75%, the foam samples were cut to a dumbbell shape with a thickness of 2 mm according to the method standard of GB/T 528-2009 in accordance with Figure 1. A universal testing machine (WDW-20D, Bairoe Test Equipment Co., Ltd., Shanghai, China) was used to apply tensile force at a speed of 500 ± 50 mm/min until the specimen broke, and the test was repeated five times under the same test conditions. The elongation at break of the specimens was calculated according to Equation (3):(3)$$E=\frac{L-L_{0}}{L}\times 100\%$$
where *E* is the elongation at break of the specimen, %; *L* is the specimen fracture distance, mm; *L*_0_ is the specimen original distance, mm.

The tensile strength of the foam samples was tested according to the national standard GB/T 528-2009. The foam samples were cut according to the dimensional requirements in Figure 1 and tested using a universal testing machine (WDW-20D, Bairoe Test Equipment Co., Ltd., Shanghai, China) with a jig stretching rate of 500 ± 50 mm/min until the specimens were pulled off, and the maximum tensile force and the cross-sectional area of the specimens were recorded and calculated according to the Equation (4):(4)$$TS=\frac{F}{A}\times 10^{3}$$
where *TS* is the tensile strength of the sample, kPa; *F* is the maximum load on the sample, N; *A* is the original cross-sectional area of the sample, mm^2^.

The tear strength of the samples was tested according to the national standard GB/T 6342-1996. The foam samples were cut into rectangular specimens, as shown in Figure 2, and each specimen should have a 50 ± 5 mm long cut on one side. The samples were tested with a tear strength tester (INSTRON-6800, INSTRON Ltd., Boston, MA, USA) with a fixture moving at a speed of 50 mm/min. The test results were calculated according to Equation (5):(5)$$R=\frac{F}{d}$$
where *R* is the tear strength, N/m; *F* is the maximum tear value, N; *d* is the initial average thickness of the formula, m.

## 3. Results and Discussion

Figure 3 shows the FTIR spectra of tannin, PVOH, TP, TFG, and TFGP. The C=C bond and aromatic group conjugate absorption peaks were found at 2300 cm^−1^. The reactive groups contained in TFG and TFGP were dominated by CH_2_-, C−O−C, and −OH. Due to the presence of absorption of multiple groups that overlap each other, the peaks at 1627, 1597, 1591, and 1513 cm^−1^ indicate the presence of a large number of −OH and C=C in furfuryl alcohol and benzene ring, which are mainly attributed to the phenolic hydroxyl groups carried by benzene ring and benzene ring. Meanwhile, the absorption peaks at 1379 and 1351 cm^−1^ are attributed to C−O−C symmetric and antisymmetric stretching vibrations, indicating that the hydroxymethyl group on furfuryl alcohol undergoes hydroxyl aldol condensation reaction with the characteristic group of glyoxal. In addition, the peak at 2843 cm^−1^ in PVOH refers to the −CHO absorption peak, which indicates that PVOH underwent isomerization reaction to form a dilute aldehyde structure, and the peak of PVOH at 2843 cm^−1^ was transferred to 2875 cm^−1^ in TP and 2932 cm^−1^ in TFGP, respectively. The FTIR test analysis revealed to some extent, the condensation reactions between tannins and polyvinyl alcohol, as well as tannins, furfuryl alcohol, and glyoxal (Figure 2).

Figure 4 shows the microscopic morphology of the four sets of samples. It can be clearly seen from the figure that the foam samples consist of open-cell structures (shown in circles) and closed-pore structures (shown in rectangles). The foam pore structure of sample T shows an irregular shape with more open pore structures, along with thinner pore walls. Since the foam samples were prepared by hand cutting, it can be observed that the foam structure on the cut surface was disrupted, and fragments remained inside the vesicles. This is due to the fact that tannin is a rigid material, and the prepared material is less flexible and easily broken. TFG foam has regular and smooth pores with few collapses between the pores compared to T foam. This phenomenon may be due to the reaction of tannins with furfuryl alcohol under the cross-linking action of glyoxal, which increases the cross-linking degree of the resin and makes the overall structure of the foam dense and homogeneous. In addition, it can be observed that there are some “small windows” (shown by arrows) on the walls of the closed-pore foam pores. After the addition of PVOH, it can be observed that the uniformity of the pores and the thickness of the pore walls of the TP and TFGP samples were improved, the number of open-cell structures and “small windows” on the pore walls was reduced, and the thicker pore walls could effectively improve the overall strength of the foams. In addition, PVOH effectively increases the toughness of the foam material so that the surface structure will not be damaged during cutting. It can be inferred that PVOH can effectively improve the bubble pore structure of foam samples and reduce the existence of bubble pore defects.

Since the size of the pores inside the foam is related to the density, the size of the pores inside the foam can be effectively inferred from the apparent density test. A larger pore size can wrap more air, which in turn, has an effect on the thermal conductivity of the foam. The results of the apparent density and thermal conductivity of the foam are shown in Figure 5. The greater the apparent density of the foam, the smaller its internal pore size [32]. The density of TFG foam (79.73 kg/m^3^) is greater than that of T foam (73.25 kg/m^3^), which proves that the cross-linking reaction of tannin and furfuryl alcohol forms a dense reticulation, which results in a denser pore structure of the foam. This phenomenon can be corroborated by the SEM images. The size of the pores inside the T foam is larger than that of the TFG foam, and the number of open-cell structures in the T-foam is higher, so the density of the T foam is smaller than that of the TFG foam. It can also be observed that the addition of PVOH further increases the densification of the internal pores of the foam, resulting in a smaller pore size and an increase in the thickness of the wall of the pores, and hence an increase in the apparent density of the foam. The thermal conductivity test results were consistent with the apparent density test results, and the thermal conductivity of TFGP foam was 0.0315 W·m^−1^·K^−1^, which was slightly higher than that of TFG foam (0.0295 W·m^−1^·K^−1^). From the SEM images, it can be clearly observed that the addition of toughening agent PVOH made the foam's internal pore structure denser and increased the thickness of the pore wall so that the air trapped inside the foam was reduced and the thermal conductivity increased. However, compared with ordinary polyurethane foam insoles, TFGP also achieved a suitable thermal conductivity [33].

The relationship between repeated foot-ground impact and overuse injuries in sports makes shock absorption performance a key design feature of insoles. Therefore, the hardness of the insole is a key factor affecting the shock absorption performance of the footwear [34]. The Shore hardness of the foam samples is shown in Figure 6. TFG and T samples are brittle materials, which is due to the large number of benzene rings in the tannin structure, and the resulting foam samples are brittle, and the data tested are affected by the depression of the sample surface under the pressure of the hardness tester probe (as shown in the circle in Figure 6b). In contrast, samples with the addition of the toughening agent PVOH, such as TFGP and TP, showed no indentation of the probe on the sample surface after the test. This is mainly due to the cross-linking reaction between tannin and furfuryl alcohol to form a network-loaded structure, while the addition of toughening agent effectively enhances the closed-pore structure and the toughness of the samples, which improves the overall performance of the samples. The structural integrity can be maintained under the pressure of the hardness tester probe and no collapse occurs. The test results of the hardness of TFGP and TP were 42 and 47, respectively. The common EVA foam insole on the market has a hardness of 48 [35]; therefore, the hardness of TFGP foam as an insole material meets the use requirements.

The results of the rebound degree and elongation at break tests of the foam samples are shown in Figure 7a. Since the TFG and T foams are rigid foams with no elasticity, the falling steel balls will break the samples directly during the resilience test, and there is no rebound phenomenon. The rebound degree of TFGP and TP were 6.3% and 5.7%, respectively. The addition of PVOH effectively improved the elasticity of the samples, which could maintain their integrity and bounce the balls up to a certain height when the samples came into contact with the falling steel balls. At the same time, the tight pore structure of the foam can deform and maintain the structural integrity when it is subjected to force (Figure 6b), so the samples have good resilience characteristics and elongation at break. Although the thermal conductivity of TFGP is slightly higher than that of TFG foam (Shown in Figure 4), its own flexibility is higher than that of TFG foam. 

Figure 8 shows the tear strength and tensile strength of this foam sample. Under stress conditions, small cracks or fissures in the material will accelerate the expansion of cracking until destruction, a phenomenon known as the tearing of the material. The tearing strength of T, TP, TFG, and TFGP is 0.32, 0.86, 0.55, and 1.43 kN/m^−1^, respectively. T and TFG foams, due to the rigidity of the material, the material will crack rapidly when subjected to external forces, and with the increase in force, the material will accelerate the expansion of cracking until destruction. However, the tear strength of TFG foam is higher than that of T foam due to the denser internal network structure. The addition of PVOH can improve the pore structure of the foam on the one hand, which makes it difficult for the sample to crack when subjected to external force, and on the other hand, the toughness of PVOH can reduce the damage of the sample caused by cracks.

It can be clearly seen that the addition of PVOH in the foam is beneficial to improve the tear strength and tensile strength compared to TFG and T. In particular, the addition of PVOH made the tear strength of TFGP (1.43 kN/m^−1^) higher than that of TP (0.86 kN/m^−1^), which depends on the cross-linking reaction between tannin and furfuryl alcohol to form a network-like structure of the foam and improve the tear strength of the foam, which was confirmed by FT-IR and some references [16]. Meanwhile, the addition of PVOH made the TFG and T samples softer while improving the network-like structure inside the foam and enhancing the tensile properties of the foam. Importantly, the tensile strength of TFGP (0.93 MPa) was more than three times that of the TP foam (0.32 MPa), and the difference in tensile strength could also be confirmed by the cross-linking reaction of tannin and furfuryl alcohol.

## 4. Conclusions

Polyvinyl alcohol-based composite foams can be prepared in an oven by a simple process while containing components from forestry resources such as tannins and furfuryl alcohol from poplar trees. The produced bio-based foams are considered meaningful due to their good appearance and absence of collapse and breakage. The physical mixing of the tannin resin and PVOH took place at the same time, while the reasonable compatibility of PVOH with the tannin resin was responsible for its uniform dispersion in the resin matrix. Compared with the original TFG foam, TFGP foam shows better toughness and has good insulation properties. In addition, SEM analysis shows that glyoxal can be used as a cross-linking agent to improve the cross-linking reaction of tannin and furfuryl alcohol, and the internal foam pores of the prepared foams are smooth and uniform without the presence of collapse so that the foams have a good reticulation structure. The addition of PVOH as a toughening agent increased the number of closed-cell structures and made the foam's internal vesicle structure denser, which improved the overall mechanical properties of the foam to a certain extent, thus making the tear strength and tensile strength of TFGP foam better than that of TFG and T foam. In the near future, based on its good development prospect, TFGP foam can be considered as a biomass foam to replace petrochemical foam and be widely used in the field of sports products.

## Data Availability

Not applicable.

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
