# Peer review of "Preparation and Characterization of Biomass Tannin-Based Flexible Foam Insoles for Athletes"

_polymers, 2023, doi:10.3390/polym15163480_

Round 1

Reviewer 1 Report

After reviewing the article "Preparation and characterization of biomass tannin-based flexible foam insoles for athletes", I am ready to express support for this work. In this work, the materials are presented sequentially and the description looks quite complete.

The paper proposes the use of tannins and furfuryl alcohol obtained from forest resources for the preparation of tannin-furan foams. To strengthen the material, the use of polyvinyl alcohol was considered, which made it possible to obtain a flexible mesh structure that gives foams excellent flexibility with the acquisition of relative properties, even close to the properties of polyurethane foams, which are the most commonly used polymers for the manufacture of insoles for athletes. This is due to the crosslinking reaction of tannin and furfuryl alcohol, which gives the foam a strong mesh structure, while the addition of polyvinyl alcohol also improves the overall mechanical properties of the foam, which increased the tear and tensile strength for such foam compared to tannin-furfuryl alcohol-glyoxal foam without the use of polyvinyl alcohol. Such material has development prospects for use in the field of sporting goods.

Despite the obvious advantages of the work, some questions are not well-defined:

1.                 The increasing consumption of shoes will lead to an increase in polymer waste in our environment. What kind of biodegradation can tannin-based foam samples have?

2.                 Is it possible to use the specified preparation process for the manufacture of large samples, which is required when creating insoles for athletes?

Author Response

Comments and Suggestions for Authors:

After reviewing the article "Preparation and characterization of biomass tannin-based flexible foam insoles for athletes", I am ready to express support for this work. In this work, the materials are presented sequentially and the description looks quite complete.

The paper proposes the use of tannins and furfuryl alcohol obtained from forest resources for the preparation of tannin-furan foams. To strengthen the material, the use of polyvinyl alcohol was considered, which made it possible to obtain a flexible mesh structure that gives foams excellent flexibility with the acquisition of relative properties, even close to the properties of polyurethane foams, which are the most commonly used polymers for the manufacture of insoles for athletes. This is due to the crosslinking reaction of tannin and furfuryl alcohol, which gives the foam a strong mesh structure, while the addition of polyvinyl alcohol also improves the overall mechanical properties of the foam, which increased the tear and tensile strength for such foam compared to tannin-furfuryl alcohol-glyoxal foam without the use of polyvinyl alcohol. Such material has development prospects for use in the field of sporting goods.

Despite the obvious advantages of the work, some questions are not well-defined:

  1. The increasing consumption of shoes will lead to an increase in polymer waste in our environment. What kind of biodegradation can tannin-based foam samples have?

Response: Thank you for the reviewer's feedback. First, the main raw materials used to prepare tannin foams are tannin, furfuryl alcohol and PVOH, all of which are biodegradable. Secondly, our group tested the biodegradation of tannin resin-based foams in a previous study using Penicillium sp. We found that there was a 0.68% weight loss of the foams after 30 d. Moreover, the SEM test clearly observed the growth of the colonies inside the foams (DOI: https://doi.org/10.3390/polym14235140). In conclusion, tannin resin-based foams are biodegradable materials and their waste does not pollute the environment.

  1. Is it possible to use the specified preparation process for the manufacture of large samples, which is required when creating insoles for athletes?

Response: Thank you for the reviewer's feedback. When making insoles for athletes, a large number of samples can be produced using a specified preparation process. In general, the process of preparing tannin resin-based foams is very simple. First, the tannin resin can be prepared in large quantities according to the proportions of the recipe, then the emulsifier and blowing agent are added to the resin and stirred, and finally the mixture is cured at 60°C. The process can be modified to produce large quantities. The modification of the preparation process can be carried out in an assembly line, and a large number of samples can be produced.

Reviewer 2 Report

The paper is discussing the use of natural biomass materials as a substitute for petroleum-based materials in the preparation of insoles for athletes. The introduction provides a brief overview of the global footwear production and the need for biodegradable footwear components. The text then goes on to discuss the use of condensed tannin-furfuryl alcohol co-condensation resin-based foams as a potential material for insoles, and the use of polyvinyl alcohol (PVOH) as a toughening agent to improve the properties of the foam.

Overall, the text is well-written and provides a clear and concise overview of the topic. The use of citations and references adds credibility to the information presented. However, it would be helpful to include more information about the specific methods used to prepare the foam and the results obtained from testing its properties. Additionally, it would be useful to provide more context about the significance of this research.

Author Response

Comments and Suggestions for Authors:

The paper is discussing the use of natural biomass materials as a substitute for petroleum-based materials in the preparation of insoles for athletes. The introduction provides a brief overview of the global footwear production and the need for biodegradable footwear components.

The text then goes on to discuss the use of condensed tannin-furfuryl alcohol co-condensation resin-based foams as a potential material for insoles, and the use of polyvinyl alcohol (PVOH) as a toughening agent to improve the properties of the foam.

Overall, the text is well-written and provides a clear and concise overview of the topic. The use of citations and references adds credibility to the information presented. However, it would be helpful to include more information about the specific methods used to prepare the foam and the results obtained from testing its properties. Additionally, it would be useful to provide more context about the significance of this research.

Response: We are very grateful to the reviewers for their valuable suggestions. A total of four sets of samples were prepared for this experiment. T foam was prepared by mixing tannin (T) with distilled water, adding emulsifier Tween 80 and guar gum, mixing well and placing in an oven for curing. TP foam was obtained by additionally adding PVOH to the preparation of T foam. TFG foam was prepared by reacting tannin (T), furfuryl alcohol (F) and glyoxal (G) to obtain TFG resin, which was cured at the same temperature after adding emulsifier and foaming agent to the resin. Finally, PVOH was added to the preparation of TFG foam to obtain TFGP foam. The context about the significance of this research has been added in the “Introduction” section, with the additions highlighted in yellow.
